# Precision Medicine—Are We There Yet? A Narrative Review of Precision Medicine’s Applicability in Primary Care

**DOI:** 10.3390/jpm14040418

**Published:** 2024-04-15

**Authors:** William Evans, Eric M. Meslin, Joe Kai, Nadeem Qureshi

**Affiliations:** 1Primary Care Stratified Medicine (PRISM), Division of Primary Care, University of Nottingham, Nottingham NG7 2RD, UK; joe.kai@nottingham.ac.uk (J.K.); nadeem.qureshi@nottingham.ac.uk (N.Q.); 2PHG Foundation, Cambridge University, Cambridge CB1 8RN, UK; eric.meslin@outlook.com; 3Dalla Lana School of Public Health, University of Toronto, Toronto, ON M5S 1A8, Canada

**Keywords:** precision medicine, personalised medicine, primary care, genomics, pharmacogenomics, prediction modelling, artificial intelligence

## Abstract

Precision medicine (PM), also termed stratified, individualised, targeted, or personalised medicine, embraces a rapidly expanding area of research, knowledge, and practice. It brings together two emerging health technologies to deliver better individualised care: the many “-omics” arising from increased capacity to understand the human genome and “big data” and data analytics, including artificial intelligence (AI). PM has the potential to transform an individual’s health, moving from population-based disease prevention to more personalised management. There is however a tension between the two, with a real risk that this will exacerbate health inequalities and divert funds and attention from basic healthcare requirements leading to worse health outcomes for many. All areas of medicine should consider how this will affect their practice, with PM now strongly encouraged and supported by government initiatives and research funding. In this review, we discuss examples of PM in current practice and its emerging applications in primary care, such as clinical prediction tools that incorporate genomic markers and pharmacogenomic testing. We look towards potential future applications and consider some key questions for PM, including evidence of its real-world impact, its affordability, the risk of exacerbating health inequalities, and the computational and storage challenges of applying PM technologies at scale.

## 1. Introduction

“Variability is the law of life, and as no two faces are the same, so no two bodies are alike, and no two individuals react alike and behave alike under the abnormal conditions which we know as disease” [1].

Precision medicine (PM) is a popular though poorly defined term in the academic and general health literature [2]. It is often used interchangeably with targeted; stratified; individualised; and personalised medicine. Personalised medicine is perhaps the most ambiguous of these terms, sometimes used to describe holistic or integrative medicine [3], or when used to describe PM to also capture concepts related to an individual’s beliefs, attitudes, and preferences towards healthcare [4], concepts central to providing holistic patient care. To avoid semantic ambiguity, we will use the term precision medicine (PM) in this review, as defined by the US National Academy of Sciences:

“the use of genomic, epigenomic, exposure and other data to define individual patterns of disease, potentially leading to better individual treatment” [5].

The promise of PM is the optimisation of the care of patients. By screening and surveillance based on better risk stratification, earlier recognition of disease, a molecularly defined diagnosis, and a more accurate diagnosis, the most effective treatment for that individual can be delivered at the best time for the most optimum outcome [4].

The growth and diversity of PM publications and initiatives in recent years have been largely driven by the advances in two technologies: molecular biology, especially the reduction in cost and the expansion of the use of genomic sequencing, and advances in computational approaches for big data analysis. So-called -omic technologies such as whole genome sequencing (WGS) generate vast amounts of data, and the use of electronic health records (EHRs) as well as wearable monitoring devices provide a further source of rich biomedical data [4,6,7]. Government-driven PM initiatives such as the “All of Us” research programme in the US [8], across the EU [9], and more widely [10] have been largely genomics-focused. It has been estimated that by the end of 2025, 52 million genomes will have been sequenced across the globe, most of which will have been performed in North America and Europe, with 40.5 million, with also large genomic sequencing initiatives in Asia but relatively few in Africa, with 42,000 [11].

These advances have given momentum to many PM developments and the much heralded promise that PM will lead to a paradigm shift in health across one’s lifetime [4,8,12]. This enthusiasm is tempered by a healthy degree of scepticism, sometimes expressed by the same authors. Some comment that PM is over-hyped [13,14] and that this focus of attention and resource on the individual presents a tension with delivering affordable and equitable population medicine, exacerbating health inequality. 

In this review, we focus on primary care and the non-specialist setting. We examine current examples of PM and its emerging applications and the opportunities and challenges PM presents with an eye on the future; this review considers PM in the context of -omics and big data (Figure 1). We have included electronic health records (EHRs) and wearable technologies within big data as both are highly relevant to primary care and crucial to capturing the data that enable PM.

## 2. The Emergence of -Omics

### 2.1. Genomics

Genomic medicine, distinct from traditional genetics, is the study of the function and interaction of the entire genome [15]. Until recently, the clinical application of genomics has been largely restricted to cancer, rare inherited diseases, and pharmacogenomics testing for oncology [16]. However, as genomic knowledge increases and the cost of sequencing falls, the frequency of use and breadth of applications has increased. A wide range of projects to mainstream genomic technologies have emerged, with countries competing to lead the way in incorporating these technologies into routine care [4,12,17,18]. 

There are three PM genomic applications that may impact primary care: (1) refined molecular diagnoses, (2) polygenic risk scores, and (3) pharmacogenomics.

#### 2.1.1. Molecular Disease Definition

Genomic and other molecular technological advances have led to a new and evolving molecular taxonomy of disease [4]. This taxonomy reflects a greater recognition of the molecular basis of disease with the identification that many diseases that were previously considered a single entity are in fact made of multiple “subtypes”, a product of both the complexity of the disease pathology and the physiology of the individuals affected [18]. Defining diseases into subtypes by genomic and other biomarkers is now standard practice in oncology, enabling targeted treatments with a significant impact on outcomes [19]. Applications in other clinical areas are more limited, although emerging examples exist in cardiovascular disease, neurology, diabetes, and autoimmune and respiratory diseases [20]. For example, asthma is now viewed as a syndrome encompassing several distinct yet inter-related diseases, each driven by a unique set of genetic and non-genetic risk factors [21,22]. As the understanding of the molecular basis of each asthma subtype is enhanced, so will the clinical models improve and with this the interventions to prevent asthma, predict the disease course, and manage the disease [23]. 

#### 2.1.2. Polygenic Risk Scores

Genome-wide association studies (GWASs) have led to the development of polygenic risk scores (PRSs), based on the aggregate effect of multiple or even thousands of variants across a person’s genome. Unlike monogenic disease risk, such as *BRCA1*, gene variants for breast and ovarian cancer, or familial hypercholesterolemia for cardiovascular disease, these variants may only confer a modest effect but, when combined, lead to a clinically meaningful and actionable level of risk. For example, the eMERGE study is generating genome-informed risk assessments (GIRAs) for 11 conditions. The GIRA incorporates the PRS, monogenic risk where applicable, family history, and other clinical factors to generate a summary risk report [24].

Although disease areas with PRS development are growing, examples in routine practice are limited. Risk stratification for cancer screening is the most active area. Currently, cancer screening is typically one-size-fits-all, with screening dictated solely by age thresholds. Breast cancer screening is an exception, with family history informing the age and frequency of mammography and testing for the oncogenes BRCA1 and BRCA2 [25]. The PROCAS study, a large UK-based cohort, is exploring the role of PRSs to further refine breast cancer screening [26], and the BARCODE1 study uses a prostate cancer PRS in primary care to identify men at risk and then offer further testing [27]. In cardiovascular disease, there has been a recent UK pilot incorporating a PRS with the standard risk prediction tool QRISK2 to refine CV risk [28]. 

There are several important considerations regarding PRSs. 

Firstly, what level of refinement they offer beyond the existing methods of risk assessment. For example, previous work in cardiovascular disease showed adding PRSs made only a slight increase in (discriminatory) predictive power [29]. In cancer, especially rare cancers, PRSs often have a modest effect on absolute disease risk. A polygenic risk score for ovarian cancer gives a lifetime risk of 2.1% for those in the top 5% of risk, compared to 1.6% in the general population [30,31].

Secondly, where will the PRS fit in the clinical pathway, a decision critical for both clinical utility and cost-effectiveness. For example, in cardiovascular risk prediction, the cost-effectiveness of the PRS to guide statin prescribing is unclear [32]. It may be that rather than guiding prescribing decision making, combining the PRS with clinical data captured from the patient’s EHR may be best used to prioritise those who should have a formal cardiovascular risk assessment [33]. 

Thirdly, PRSs may risk exacerbating health inequality. PRS performance is not always maintained across genders and those of non-European ancestry, reflecting a lack of diversity in dataset development [24,34,35]. Implementing these biased PRSs into clinical practice risks worsening inequality. Initiatives to diversify the patient cohorts used for PRS development, such as the Global Biobank Meta-analysis Initiative (GBMI) [36], will help the broader applicability of PRSs. 

Fourthly, no matter how well optimised the PRS is, there will always be a ceiling to its accuracy, as they do not capture lifestyle risk factors nor other poorly understood non-genetic factors [30,37]. Therefore, to maximise clinical utility, PRS results will need to be combined with risk data from other sources. Examples of this approach include eMERGE [24] and BOADICEA, a breast cancer risk prediction model, that combines PRSs with monogenic risk factors, lifestyle/hormonal and reproductive risk factors, and mammography breast density measurements [38]. 

Even when optimised to maximise clinical utility, a central ethical consideration exists: What happens when you inform a person of their risk? Is there something fundamentally different in response to a PRS-informed risk rather than a risk derived from other methodologies? Concepts of genetic determinism are often poorly understood, with a tendency to exaggerate the anticipated magnitude of the impact of genetic findings [39]. This may affect individuals differently, with some feeling overly reassured by a low PRS and therefore not engaging in routine screening, which may be easy to do, cheap, and effective [37]. In contrast, those with a high PRS may then adopt a fatalistic view, thinking there is no point making the lifestyle changes as their outcome is predetermined [40]. These questions remain unanswered [37,41].

#### 2.1.3. Pharmacogenomics

Emerging from the convergence of pharmacology and genomics, pharmacogenomics (PGx) is the study of the variations in DNA and RNA that relate to drug response [42]. PGx aim to individualise drug therapy, to ensure that prescriptions are “The right drug, for the right patient at the right dose” [43]. 

Adverse drug reactions (ADRs) are common and cause significant iatrogenic harm, with ADRs estimated to affect 10% of inpatients [44], and the annual cost of avoidable ADRs in the UK is estimated at GBP530 million [45] and more than USD30 billion in the US [46]. Additionally, drugs that lack efficacy for that individual will not only negatively impact their care but also lead to additional costs for the healthcare system, with disease relapse or deterioration leading to hospital admission or an extension of inpatient stay [47]. 

Emerging examples of PGx use in primary (and other non-specialist) settings include the following:Improving drug safety. Genotyping for HLA-B*1502, highly prevalent in certain Asian ethnic groups, reduces the risk of a severe and frequent reaction to carbamazepine [48] with pre-treatment screening advised in the BNF [49].Improving drug efficacy. Variation in the cytochrome P450 gene *CYP2D6* affects the metabolism and elimination of more than 100 drugs [50]. One of these drugs is the analgesic codeine, which is metabolised to the bioactive form morphine. Patients can be classified by their rate of metabolism, with clinical implications for the ultrarapid metabolisers (UMs) and poor metabolisers [51], and pharmacogenomic guidance in the summary of product characteristics (SmPCs) [52]. Variants for another cytochrome P450 gene, *CYP2C19*, can also significantly impact clopidogrel metabolism and efficacy with an FDA “black box warning” for those carrying these variants [53].

There are a range of further uses in specialist settings that include gene variant-specific drug treatments [42], to avoid adverse drug reactions such as the routine use of *HLA-B*57:01* genotyping prior to prescribing the anti-HIV drug abacavir [54], and to guide drug dosing, such as *DPYD* gene testing for 5-fluoruracil [55].

There are now actionable drug–gene interactions for many commonly prescribed drugs in primary care, including codeine, clopidogrel, antidepressants, and statins. The PharmGKB database lists 428 drugs with a pharmacogenomic FDA label annotation,137 of which have a requirement for genomic testing [56]. 

Outside of specific targeted secondary care indications, PGx have not been widely implemented into any healthcare system [47]; however, with a growing number of indications, there are multiple initiatives to incorporate testing into routine practice. Primary care is perhaps the most suitable setting for implementing PGx clinical decision support. Most prescribing decisions are performed in primary care and there is already the widespread use of primary care EHR systems that are integrated with prescribing software [47]. 

Implementing a PGx test includes considering not just what should be tested for but also where, when, and how the test is performed and then how the result can be easily retrieved when needed. There are broadly two approaches to PGx testing: Testing at the time of prescribing. Typically, this is a single drug–gene test performed in advance of the prescription decision about to be made. For example, in oncology, *DPYD* testing in advance of initiating 5-FU, or in neonatal sepsis, testing the *RNR1* gene for variants associated with aminoglycoside-induced hearing loss [57]. Opportunities for this approach, rapid testing for a single and significant gene–drug interaction, will broaden as molecular diagnostics continue to advance [58]. Although point-of-care testing (POCT) is already utilised in primary care [59], it is hard to see PGx POCT expand beyond a relatively limited number of indications. In primary care, given the range of clinical presentations and prescribing decisions, the feasibility of POCT, and the effect of even a modest delay on patient flow, an alternative approach is likely to be better suited.Testing patients in advance of prescribing decisions. This is a more distant prospect for primary care but would involve pre-emptively performing a PGx gene panel test for several key drug–gene interactions with the results captured to inform future prescribing decisions. This may involve testing at a separate time to the prescribing decision or be triggered by prescribing a single drug on the panel, the specific drug information available to guide that treatment, and the other PGx information available for future reference. An attractive approach is to use the existing sequencing data, captured for another indication to identify PGx variants, which is increasingly feasible as an increasing proportion of the population has genome or exome sequencing [47].

Whichever approach is used, capturing PGx data is only worth doing if it is available and easily interpretable at the point of future prescribing decisions. Currently, in primary care, this is likely to mean targeted pre-emptive testing in those patients who are more likely to be prescribed specific medications, with the result recorded in the EHR and integrated with prescribing software to give actionable prompts at the point of prescribing. 

### 2.2. Transcriptomics, Epigenomics, Proteomics, Metabolomics, and Exposomics

As technology has improved, so too has the capacity to study not only the genome but also the multiple steps for modification between this genetic code and the final product, the protein. The -omics describe elements of these steps and their outcomes. Exposomics, however, is different; it is an emerging field in PM that attempts to capture the totality of chemical and non-chemical exposures (e.g., physical activity, diet, psychosocial stressors, and toxins) that occur across an individual’s lifetime [60]. 

Transcriptomics describes the sum of all the RNA transcripts, the different readings of the genome [61]. This captures the dynamic state of the cell, identifying gene fusions, posttranscriptional changes, and other RNAs (e.g., microRNA, ribosomal RNA, and transfer RNA) that play key roles in fine-tuning how the RNA transcript is read and how the protein is made. It enables a far greater insight into what is happening at a functional level than genomics. For example, an emerging area of transcriptomics research is the impact of environmental exposures on how RNA transcripts are processed and modified [62].

Epigenomic factors, meaning on top of the genome, influence what parts of the genome are read and how thoroughly. They play an important role in cancer and the predisposition and risk of complication in common diseases [61,63]. 

Proteomics is the measurement of the final product (proteins) and how they vary from cell to cell and under different influences, with metabolomics the term used when the protein affects a metabolic pathway. Proteomics shows promise in asthma patient stratification. Asthma is the consequence of a pathogenic process that involves both environmental and genetic risk factors. A proteomic analysis of body fluids (e.g., serum, sputum, or bronchiolar lavage) gives an insight into the consequences of both these risk factors and a better understanding of the biological mechanism driving the patient’s asthma with the potential for precision therapeutic targets [64].

These -omic technologies generate vast amounts of data, the processing and interpretation of which is a substantial undertaking requiring the input of expert bioinformaticians [61]. This is true with a single-omic technology but even more so when several technologies are incorporated into the analysis, sometimes referred to as the “multi-ome” or “panor-ome” [61]. AI is increasingly being deployed to manage and interpret these multi-omic “big data” [65]. In primary care, the closest multi-omic application is in cardiovascular disease prevention and management, combining proteomic and genomic markers [66].

## 3. Big Data, Data Analytics, and AI

### 3.1. Electronic Health Records

In the UK and increasingly in other countries, primary care is almost entirely paperless. The widespread use of electronic health records (EHRs) is driven by the need to capture healthcare utilisation for the purpose of reimbursement and to improve the recall of information to optimise patient care. EHR systems also provide a resource of data that can be interrogated at a population level.

Many advances in PM have utilised primary care EHR research datasets, in their development, with patient-level data used to stratify patients into subgroups for targeted clinical care and treatments [16,18]. The EHR can also act as a platform for implementation of PM in primary care, integrating clinical decision support tools. However, there remain challenges to address. 

Incorporating non-structured data. EHR data are typically held in multiple different formats: coded data, numerical values, images, and free text. Non-structured formats of data, such as free text, are of limited use, and they are often excluded from anonymised research datasets to preserve participants’ anonymity [67], and natural language processing (NLP) software historically has been of limited use. The situation is changing with significant advances in NLP [68], although many of the information governance concerns remain. 

Data need to be standardised so they are comparable and consistent. A coded term for a diagnosis or intervention in one setting should mean the same in another setting, for example, a diagnosis of type 2 diabetes is the same in all settings, in primary care and secondary care. This is a necessity for efficient data integration and interoperability and the focus of initiatives across the world [65,69].

EHR linkage. PM can be further refined by linking to other sources of patient data including other healthcare settings, such as secondary care records and radiology/pathology results and images, and non-healthcare settings: education, social care, and national statistics data, such as levels of deprivation [4]. Accurate linkage requires a data infrastructure that minimises linkage error, ideally with a clear unambiguous patient identifier across datasets, with the standardisation of coding across these settings. The product of this is a linked dataset that is not unnecessarily large and unwieldly, both for ease of use but also to fulfil information governance responsibilities regarding data minimisation.

Information governance concerns. When using healthcare data, there are sensitivities and legal restrictions on how they are used and shared. In the UK, there is a broad, but not universal, consensus for sharing data for clinical care and sharing anonymised data for research. There is far less agreement when anonymised data are shared with private companies [70], who are often at the vanguard of PM projects, with concerns regarding data use and the risks for patient confidentiality. For example, there have been major concerns in the UK about awarding the contract for the NHS federated data platform to the US company Palantir [71]. A further complicator is the role of the primary care physician or general practitioner as a data controller for UK primary care EHR data. Although the data are held by a few EHR providers, the responsibility for these data sits with thousands of practice-level decision makers. Consequently, decisions about what can and cannot be done vary significantly with different interpretations of their position and responsibilities, as demonstrated by a reluctance to release records for key PM research initiatives, such as the UK Biobank [72]. 

### 3.2. Digital Technologies including Wearables

A further rich source of data is now captured by the patient both through intermittent home measurements and wearable technologies. These wearable technologies can capture location, activity, and physiological parameters, such as blood pressure, pulse, oxygen saturation, respiratory rate, and temperature, in a near continuous manner. This enables a deeper understanding of disease, including intra- and inter-patient variability, beyond what is possible with intermittent clinic-based measurements [73]. The information can be used to both diagnose disease, for example, capturing heart arrhythmias (e.g., atrial fibrillation), and to monitor and help manage disease [74].

Wearable technologies also open the possibility to capture data on the major determinants of health, not normally captured in health data, with measurements related to lifestyle, nutrition, and the environment. Such data will be crucial if PM interventions are to make a significant impact upon broader societal health [75,76]. The storage, processing, and interpretation of these data pose a further big data challenge. 

### 3.3. Prediction Modelling 

Clinical prediction tools are the most widely used PM application in primary care. They involve modelling the relationship between future or unknown outcomes (endpoints) and baseline health states (starting points) [77]. They are well established, used across medicine for more than 40 years [78,79], with widely adopted examples in primary care to stratify patients’ risk of cardiovascular disease, osteoporosis, emergency admission, and degree of frailty [80,81,82,83]. Artificial intelligence (AI) is increasingly being used to enhance modelling, enabling a broader range of models to be developed at a greater speed [84]. AI-derived models have been shown to be superior, in some but not all examples, to those developed using standard epidemiological methods for predicting disease [85,86], estimating prognosis [87], and predicting all-cause mortality [88]. As discussed previously, the inclusion of PRSs may further enhance the precision of these tools. 

### 3.4. Artificial Intelligence (AI)

Much of the promised potential and enthusiasm for PM is due to the acceleration in advances in AI. AI is a term used to describe technologies and methods that allow for machines to exhibit intelligent behaviour, an area of research and development that has grown from the convergence of advances in computing power and the availability of vast amounts of data [89]. Significant amounts of AI research efforts have focussed on healthcare, in particular PM, with a growing number of partnerships announced between AI developers and the EHR providers, the custodians of large repositories of healthcare data [90]. 

To date, many of the AI healthcare uses have used a single data type or mode, with image-based applications, such as the assessment of radiology, dermatology, pathology, and retinal images, the most widely researched and utilised applications. In ophthalmology, retinal image analysis includes clinical applications, such as identifying diabetic retinopathy [91], and research to identify retinal changes as early markers for a range of conditions [92,93,94,95]. These image applications are already impacting their respective specialties and are likely to affect workforce requirements in the next 5–10 years [96]. These AI applications have largely utilised supervising learning, requiring inputs annotated with the correct information for the AI algorithm to learn.

Examples in primary care are limited, with early examples of skin lesion analysis [97,98], including eczema assessment [99], supporting clinical documentation and coding [100], and automating some administrative tasks [101].

The goal of AI in PM is to incorporate and analyse the multimodal data of PM, -omics, EHRs, and wearable data to generate personalised actionable and timely patient insights delivered in a manner optimised for the individual patient and their clinician. The prospect of achieving this has recently been made more realistic by the progress in generative AI, in particular Large Language Models (LLMs), such as ChatGPT [102] and BARD [103]. By using unsupervised learning, without the need to annotate data prior, LLMs can both significantly increase the amount of data used and the speed with which they learn, enabling a move from narrow-AI, developed and retrained for each use case, to general AI, which is transferrable across applications. Early examples of LLMs utilised language data only, but later versions now incorporate other modes of data and are sometimes referred to as large multimodal models (LMMs), with an early medical example the use of ChatGPT-4V(ision) to both interpret the radiology image and generate the radiology report [104]. 

Looking ahead, LMMs will be able to move beyond current multimodal PM applications, such as the analysis of genomic and EHR data [76,105], to analyse several data inputs: past medical history, biological, physiological, environmental, and socio-economic data. These collated data could be used to create a “digital twin”, upon which healthcare interventions could be modelled, with the findings of the model delivered via a virtual health assistant working alongside the clinician to give patient-specific guidance on disease prevention and management, including personalised lifestyle interventions [89]. 

For example, imagine a near-future patient encounter. For a patient with pre-diabetes, hypertension, obesity, a family history of heart disease, and a raised polygenic risk score for heart disease, the virtual assistant would advise drug interventions guided by the best available literature, optimised for that patient by previous prescribing information, their pharmacogenomic data, and other -omics data regarding their individual disease drivers. The virtual assistant would then advise and coach the patient in real time regarding lifestyle interventions based upon existing physical activity, geolocation, and nutritional data captured from wearable technologies.

Similarly, in pandemic surveillance, real-time precision risk assessments could be made based on patient-specific information, such as comorbidities, vaccination status, location data, and physiological measurements from wearable sensors, combined with data such as wastewater analysis to understand the levels of the pathogen in their community [106].

These potential applications may seem a long way off, but the hyper-evolution of AI technologies and the reduction in the proposed timescales for general AI to reach human levels of performance suggest it may arrive sooner than we would expect [107]. 

The promise of AI is mirrored by significant concerns about the societal impact of advanced generative AI [108]. Some of the broad questions to consider before implementing AI tools are also captured in existing frameworks, such as the ACCE model process for evaluating genetic testing [109,110]. First, does the tool measure what it is supposed to measure? (Analytical validity). Second, would it really make that much difference, above and beyond what we already know or do? (Clinical validity). Third, does it work in the context it is going to be deployed? (Clinical utility). And fourth, is it worth the cost? Is it acceptable? And how will it impact health inequality? (Ethico-legal and social implications).

There are some AI-specific considerations, with regulatory and ethical frameworks developed to help guide the implementation of AI in healthcare [111,112]. Firstly, the financial and ecological cost of AI applications may be higher, with an LLM search estimated to be 10 times the cost and energy consumption of a conventional text-based google search [113]. Secondly, the erosion of trust in healthcare, which may be worsened by a so-called “black box result”, with healthcare decisions made by an AI system we cannot fully interrogate [114,115]. Trust is not only necessary for AI system adoption but also to maximise compliance with any recommendations made. Thirdly, like many “big data”-derived tools, its effectiveness is dependent upon the learning dataset used to create it: “the output is only as good as the input”. If the learning dataset does not reflect the real world, e.g., by under-representing certain groups in society or being biased in some other way, the AI output will be less effective and possibly inaccurate and unsafe. The term “Health Data Poverty” describes this phenomena, where under-represented groups will least benefit from the innovations derived from the data [116]. This is especially concerning when many of the multimodal data repositories are voluntary natural history studies and genomic datasets, both of which are under-represented with people from minority or diverse social and economic backgrounds [114,117]. Even when the dataset is representative, in that it reflects national census data for age and other demographic measures, if a certain group has, for example, a higher rate of misdiagnoses, then the insights derived from the data will still be biased [118].

## 4. Discussion

Precision medicine sits at the interface of two significant technological advances, molecular biology and big data analysis. The effect on healthcare is already substantial and likely to be increasingly so. Some predict a disruptive paradigm shift in clinical practice. However, precision medicine might best be regarded more as a continuation of what is currently done, and thus a natural evolution of evidence-based medicine, reducing errors and optimising care [119].

If we consider the current and near-future applications of PM in primary care, there are three key areas: pharmacogenomics, with results seamlessly integrating with electronic health record prescribing software; the greater use of patient stratification and prediction tools that incorporate polygenic risk scores; and more precise molecular diagnoses. These are evolutions of what is done already, with prescribing support software and risk prediction tools already utilised widely. Therefore, the potential for the disruption of primary care and the doctor–patient relationship can be minimised by building on these current skills and processes. 

Looking further ahead the path is less clear. Many of the PM applications are largely aspirational with the route to implementation unclear. The two key technologies, AI and -omics, will lead to more and more sophisticated PM technologies. If they are to impact patient care at scale, they will need to be implemented in primary care. There will be technology-specific questions, but many of the challenges and considerations are shared across technologies: ensuring the technology meets the needs of primary care; that the evidence base is robust and from its intended clinical setting; that the datasets upon which the PM are derived are representative; that the technology is affordable; that it does not increase health inequalities; and minimising the erosion of trust and holistic care. There are key gaps in what we know regarding these challenges; to capture this evidence before implementing PM in primary care, it is important to consider the following:**Co-development of technologies**. To date, healthcare AI tools have often been driven by a focus on the technology and commercial need to find a marker rather than patient and clinician need [101]. The co-development of PM technologies with primary care clinicians, patients, and the public is key to ensure they address the needs and meet the standards and values of society and primary care. With a specific focus on ensuring an evidence base in the primary care setting they are to be deployed, consideration should be given to their impact on continuity of care and how they fit into the consultation and how they will avoid overmedicalisation and increasing health anxiety [101]. It will also be important to ensure that the outputs of PM technologies are delivered in such a way to have optimal impact, effectively inform clinical decision making and create meaningful change in people’s behaviour whilst not excessively burdening a healthcare system already under pressure.**Real-world evidence in the population and setting it is to be deployed.** Unrepresentative and biased datasets lead to PM tools that may exacerbate health inequalities. Before implementing PM at scale, health strategies need to ensure that the foundations upon which PM is built, datasets, genetic databases, cohort studies, and EHR datasets, are appropriately diverse and representative of their intended use population. Endeavours such as the STANDING Together initiative will be key to encourage representativeness in datasets and ensure transparency in how diversity is reported [120].Fundamentally, evidence needs to be gathered on PM technology in the setting that it is intended to be deployed. Environmental factors and where the technology is placed in the current workflow can significantly impact its performance and utility [76,121]. Before the widespread adoption of PM, implementation models need to be used suitably for these new technologies [122] and appropriate evaluation frameworks applied to ensure robust real-world evidence [109,123]. To minimise the workload impact of these technologies, care will be necessary to ensure they are implemented efficiently, but how we define and then measure efficiency is not clear, especially in the context of new technologies, and is an area for further research [124].Currently, in primary care, the EHR system plays a key role in the doctor–patient consultation, not only to inform the doctor and record clinical information but to facilitate patient involvement in the consultation by sharing the monitor screen [125]. When implementing PM technologies, not only should one consider how to use the contents of the EHR to develop the precision medicine insight but also how the EHR system interface will enable the clinician and patient to best understand and implement these PM insights meaningfully. **Demonstrating the cost-effectiveness of PM.** There is still much uncertainty about the affordability and health economic profile across the range of PM interventions [126]. PM enthusiasts, and frequently policy makers, highlight the potential cost savings of PM: avoiding ill health; promoting health prevention; streamlining diagnostic pathways with earlier diagnoses; making better therapeutic decisions, with less associated waste and adverse events; and decreasing the disease burden for the public at large [76,127,128]. However, the evidence for these health system efficiencies is hard to capture with the PM interventions adding an upfront cost, with the later benefit difficult to measure. For example, currently, drugs are prescribed without pharmacogenomic information. Will a net reduction in adverse events and inappropriate prescribing, and thus in health impact and cost, justify the expenditure of pharmacogenomic testing at a population level? Capturing sufficient information to understand cost-effectiveness, clinical impact, and the long-term viability of PM is likely to require an extended period of surveillance. Such ongoing surveillance of PM interventions should be ensured from the outset, adapting existing approaches of post-market surveillance for new drugs and medical devices.Whilst PM does not in itself seek to establish novel medications, it does stratify patients into subgroups who will respond to specific treatments regimes. Molecular disease definitions divide common conditions into multiple distinct subgroups, many of which will have their own treatment. In cancer, this has led to the development of drugs that have in many settings been prohibitively expensive [129]. Repositioning affordable licensed drugs based on specific molecular targets is an attractive proposition to reduce drug costs whilst maximising efficacy [121]. Although this has not been widely utilised to date, pharmaceutical companies and government research funders could use this opportunity to revisit “old drugs” for targeted personalised therapy in specific subgroups [130]. **Data collection sharing and transfer.** Much of the potential of PM is dependent upon processing and analysing large amounts of data. Genomic sequencing has advanced at pace; however, the availability of information on diverse well-phenotyped individuals has not kept pace, hampering the ability to establish connections between disease and genomics [131]. Optimising the quality of data recorded is key. To achieve this, establishing standards for data recording, including clinical vocabulary that is used across care settings, and frameworks to share data, compliant with legal restrictions, that maintain patient privacy, and incorporating individual preferences for their data use need to be prioritised.In UK primary care, for example, better guidance regarding data sharing is needed to ensure practice is more uniform, with recent proposals that data controller responsibilities could be shared with national bodies [132]. In addition to ensuring the quality of the data recorded and standards for sharing, significant attention should be focussed on the storage and processing of the vast quantity of data that PM needs and will generate. This risks overwhelming an already stretched health system and workforce. Although plans are in place to advance NHS digital systems [133], this needs to be prioritised with suitable cautions given that previous large-scale IT infrastructure projects across the NHS have failed [134]. **Impact upon holistic care**. There is concern that the advance of PM interventions may reduce the need for human interaction, with the virtual clinical assistant taking a greater role and affecting the patient–doctor relationship. As healthcare interventions become more personalised, derived from increasingly complex methodologies, the rational for the intervention may become more opaque [114,115]. Will this lack of transparency erode trust and further impact the doctor–patient relationship? Advocates of PM suggest that it will enable the better use or resources, bringing together disparate information to support the clinician to make the best decision, thus freeing time for human intelligence and restoring empathy [135]. However, evidence for or against this position needs to be established, with robust primary care-based qualitative research incorporating the views of clinicians and patients.

## 5. Conclusions

Advances in PM will undoubtedly impact primary care, and the three key areas that will be impacted first are likely to be pharmacogenomics; polygenic risk scores informing clinical prediction tools; and precision molecular diagnoses. 

Primary care-based research should be at the centre of these advances to ensure robust evidence for their use and approach to implementation. 

This research needs to demonstrate that PM improves health outcomes for all and does not exacerbate health inequalities. That the additional insights gained from the PM intervention have a clinically meaningful difference above and beyond current practice and, importantly, that PM does not divert limited health resources to the individual at the expense of population-based healthcare interventions for the many. 

That PM is cost-effective, with suitable continued evaluation to ensure the full impact of the intervention is captured. 

That PM is acceptable to society. The need to link multiple datasets to develop increasingly granular insights also raises issues around data governance and confidentiality—how this data is used and where it is held. If the process by which these insights are derived is opaque due to the complexity of the algorithm or AI used, how will this be accepted by patients and clinicians? And how will this impact their compliance with the proposed recommendations?

If these challenges are not addressed, PM risks becoming a shiny new initiative, available only to the wealthy minority, exacerbating health inequality and diverting valuable funds from core services, with little health gain. 

## 6. Take-Home Points

Precision medicine (PM), a term often used interchangeably with targeted, stratified, individualised, and personalised medicine, is a rapidly developing area of research and practice.PM can be considered to involve two broad areas, each with its own sub-domains. (1) The “-omics”, with a whole array of prefixes including but not limited to genomics, pharmacogenomics, transcriptomics, epigenomics, proteomics, metabolomics, and exposomics. (2) Big data, data analytics, and artificial intelligence (AI).PM is data science-driven and is built upon large volumes of biomedical data. AI is a key tool to manage, analyse, and communicate the insights from the multiple big data healthcare sources.The three -omics likely to impact primary care are polygenic risk scores to optimise patient risk stratification, pharmacogenomics to tailor treatment, and molecular genetic diagnostic testing. The impact of all will be optimised when integrated with the primary care electronic health record (EHR), which forms the key data resource for PM development and implementation in this setting.The evidence base for PM is still emerging. The pressures and incentives for the early adoption of PM technologies are multifactorial and complex. There is a need for PM initiatives that have real-world evidence of clinical utility in the context they are to be deployed. Particular attention to demonstrating cost-effectiveness and effect on health inequalities is required.

## Figures and Tables

**Figure 1 jpm-14-00418-f001:**
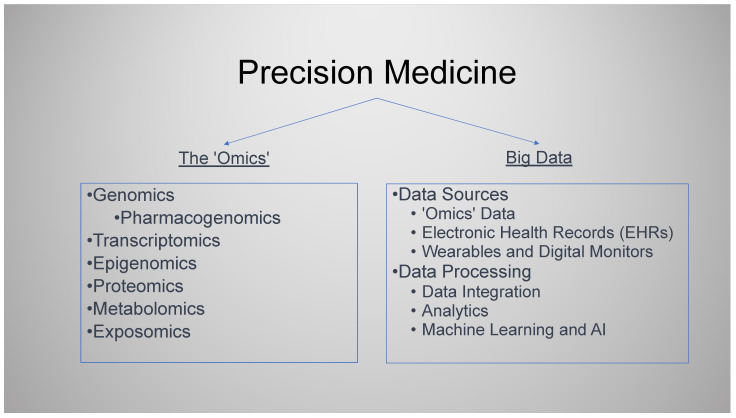
Precision medicine.

## Data Availability

Not applicable.

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
