# Peer review of "Precision Medicine—Are We There Yet? A Narrative Review of Precision Medicine’s Applicability in Primary Care"

_jpm, 2024, doi:10.3390/jpm14040418_

Round 1

Reviewer 1 Report

Comments and Suggestions for Authors

Thank you for the opportunity to read the article entitled "Precision Medicine - are we there yet? A narrative review of Precision Medicine’s applicability in primary care," which I found extremely interesting, appropriate, and relevant to the field of precision medicine.

However, I identified some aspects that could enhance the quality and coherence of the text:

1.     First, I noticed that lines 31-33 and 56-57 seem to repeat the same information.

2.     Additionally, in lines 177-180, I believe it would be beneficial to expand the discussion on how the lack of efficacy of certain medications can negatively impact healthcare services. For example, the relapses involving hospitalization and associated costs could be mentioned, further underscoring the importance of pharmacogenomics in optimizing treatment.

3.     Regarding lines 198-203, I suggest revising the paragraph to avoid conveying the impression that pharmacogenomics is not yet implemented. Furthermore, I suggest adding more specific examples, such as the mandatory pharmacogenetic analysis before administering medications like irinotecan, 5-FU, and capecitabine in cancer treatment or abacavir in HIV.

4.      I would recommend converting Figures 2 and 3 into text within the body of the article, as they seem to serve no specific function as independent figures.

5.     I believe it would be beneficial to include a final conclusion that synthesizes the key findings of the article and highlights the practical implications and future research directions in the field of precision medicine.

Author Response

1. First, I noticed that lines 31-33 and 56-57 seem to repeat the same information.

I have moved take home points to the end of the article. Then the repetition of statements is not so glaring.

2. Additionally, in lines 177-180, I believe it would be beneficial to expand the discussion on how the lack of efficacy of certain medications can negatively impact healthcare services. For example, the relapses involving hospitalization and associated costs could be mentioned, further underscoring the importance of pharmacogenomics in optimizing treatment.

I have added the following:

"Additionally, drugs that lack efficacy for that individual will not  only negatively impact their care but also lead to additional costs for the healthcare system, with disease relapse or deterioration leading to hospital admission or extension of inpatient stay(47)."

3. Regarding lines 198-203, I suggest revising the paragraph to avoid conveying the impression that pharmacogenomics is not yet implemented. Furthermore, I suggest adding more specific examples, such as the mandatory pharmacogenetic analysis before administering medications like irinotecan, 5-FU, and capecitabine in cancer treatment or abacavir in HIV.

I have rephrased the paragraph as suggested:

"Outside of specific targeted secondary care indications, PGx has not been widely implemented into any health care system (44), however with a growing number of indications there are multiple initiatives to incorporate testing into routine practice."

The focus of the article is primary care so the focus has been drugs prescribed in primary care. However I have added a couple of secondary care examples:

"With a range of further uses in specialist settings that include gene variant specific drug treatments (42), to avoid adverse drug reactions such as the routine use of HLA-B*57:01 genotyping prior to prescribing the anti-HIV drug abacavir (54), and to guide drug dosing, such as DPYD gene testing for 5-fluoruracil (55)."

4. I would recommend converting Figures 2 and 3 into text within the body of the article, as they seem to serve no specific function as independent figures.

Removed figures and incorporated into the text as suggested. See line 377-385 and 397-401.

5. I believe it would be beneficial to include a final conclusion that synthesizes the key findings of the article and highlights the practical implications and future research directions in the field of precision medicine.

I have added a conclusion after the discussion to bring together the key findings, practical implications and research of PM in primary care.

Reviewer 2 Report

Comments and Suggestions for Authors

Thank you for the opportunity to review this manuscript. The manuscript, a review paper “Precision Medicine - are we there yet? A narrative review of Precision Medicine’s applicability in primary care” presents an important topic. I have pointed out and suggested a couple of points for consideration: listed below as suggestions to be addressed.

The history and updated statistics about precision medicine at country level or regional level will be of importance for the readers to grasp the topic.

2.     The authors discuss the potential impact of precision medicine on primary health care. However, I would suggest describing it in the perspective of potential ethical issue of resource allocation, resources imbalance an health inequalities?

3.     Electronic health records are useful resources. However, linkages and data transfers are grey areas and need further discussion. It will be useful to discuss risk benefit ratio in terms of use of electronic health records for precision medicine.

Author Response

1. The history and updated statistics about precision medicine at country level or regional level will be of importance for the readers to grasp the topic.

As precision medicine encompasses several areas of practice, county and regional statistics are limited. However, I have added the following line 76 to give a more global perspective.

"Government driven PM initiatives such as the ‘All of Us’ research programme in the US (8), across the EU (9) and more widely (10), have been largely genomics focused. It has been estimated that by the end of 2025, 52 million genomes will have been sequenced across the globe. Most of which will have been performed in North America and Europe, 40.5 million, with also large genomic sequencing initiatives in Asia, but relatively few in Africa, 42,000 (11)."

2. The authors discuss the potential impact of precision medicine on primary health care. However, I would suggest describing it in the perspective of potential ethical issue of resource allocation, resources imbalance an health inequalities?

The impact of PM on resource allocation and health inequalities is highlighted in the abstract line 18, and 25. Line 82 of the introduction. In the context of biased data sets to exacerbate health inequality in PRS and efforts to improve this line 149-154.

In the discussion impact of PM on health inequalities is discussed line 45. Then considerations to minimize this risk line 466. Consideration of cost effectiveness of PGx is also included in the discussion.

I have added a conclusion after the discussion that emphasizes the impact on resource allocation and health inequality.

PM impact on workload in primary are has been expanded. Line 467

"It will also be important to ensure that the outputs of PM technologies are delivered in such a way to have optimal impact, effectively inform clinical decision making, create meaningful change in people’s behaviour whilst not excessively burdening a health care system already under pressure."

And line 485

"To minimize workload impact of these technologies, care will be necessary to ensure they are implemented efficiently, but how we define and then measure efficiency is not clear especially in the context of new technologies and an area for further research (120)"

3. Electronic health records are useful resources. However, linkages and data transfers are grey areas and need further discussion. It will be useful to discuss risk benefit ratio in terms of use of electronic health records for precision medicine.

Added see line 304

"Accurate linkage requires a data infrastructure that minimizes linkage error, ideally with a clear unambiguous patient identifier across datasets, with standardization of coding across these settings. The product of which is a linked dataset that is not unnecessarily large and unwieldly, both for ease of use but also to fulfil information governance responsibilities regarding data minimization."

Added to the discussion line 490

"Currently in primary care the EHR system plays a key role in the doctor patient consultation, not only to inform the doctor and record clinical information but by sharing the monitor screen facilitate patient involvement in the consultation (125). When implementing PM technologies not only should one consider how to use the contents of the EHR to develop the precision medicine insight, but also how the EHR system interface will enable the clinician and patient to best understand and implement meaningfully these PM insights. " 

Reviewer 3 Report

Comments and Suggestions for Authors

The manuscript entitled “Precision Medicine - are we there yet? A narrative review of Precision Medicine’s applicability in primary care” provides a comprehensive overview of precision medicine (PM) and its potential applications in primary care. This manuscript discusses the potential for PM therapies to worsen health inequity or undermine trust in the patient-doctor relationship. Further elaborating on these ideas and exploring the ethical and social ramifications of PM in primary care would enhance the depth of the discussion. Examining issues related to access, equity, confidentiality, and patient self-governance could offer significant knowledge for researchers and policymakers. Although the paper emphasizes the potential advantages of PM, it is essential to emphasize the significance of continuous assessment and long-term viability. Authors could explore approaches for conducting extended surveillance of PM therapies in primary care, encompassing techniques for evaluating efficacy, cost-effectiveness, and societal impacts. Also, take home points should be written at the end of the manuscript.

Author Response

1) The manuscript entitled “Precision Medicine - are we there yet? A narrative review of Precision Medicine’s applicability in primary care” provides a comprehensive overview of precision medicine (PM) and its potential applications in primary care. This manuscript discusses the potential for PM therapies to worsen health inequity or undermine trust in the patient-doctor relationship. Further elaborating on these ideas and exploring the ethical and social ramifications of PM in primary care would enhance the depth of the discussion. 

"I have added a conclusion that further explores this."

2) Examining issues related to access, equity, confidentiality, and patient self-governance could offer significant knowledge for researchers and policymakers.

Equity is discussed in the PRS section line 155 and discussion lien 457, 473 and revisited in the conclusion.

Confidentiality I have added a further detail related to confidentiality concerns with data linkage and sharing in line 312.

"There is far less agreement when anonymised data is shared with private companies (70), who are often at the vanguard of PM projects, with concerns regarding data use and the risks for patient confidentiality."

This is also explored in the conclusion.

Concepts around self governance are touched on in the final section of the PRS section, with concepts of patient action with polygenic risk advice. This is then further explored in the conclusion.

3)Although the paper emphasizes the potential advantages of PM, it is essential to emphasize the significance of continuous assessment and long-term viability. Authors could explore approaches for conducting extended surveillance of PM therapies in primary care, encompassing techniques for evaluating efficacy, cost-effectiveness, and societal impacts.

Added to discussion line 507

Capturing sufficient information to understand cost effectiveness, clinical impact and the long-term viability of PM is likely to require an extended period of surveillance. Such ongoing surveillance of PM interventions should be ensured from the outset, adapting existing approaches of post-market surveillance for new drugs and medical devices.

And then further discussed in the conclusion.

4) Also, take home points should be written at the end of the manuscript.

I have moved take home points to the end of the article.